# OpenReview forum: "Moloch's Bargain: Emergent Misalignment When LLMs Compete for Audience Success"
_ICLR.cc/2026/Conference — ICLR 2026 Conference Withdrawn Submission_

### Official Review · Reviewer_gYK6 · 2025-10-23

**Soundness:** 2
**Presentation:** 4
**Contribution:** 2
**Rating:** 2
**Confidence:** 4

**Summary:**

This paper performs a simulation study in which an LLM (Llama or Qwen) is used to write text to persuade an audience of LLM-simulated voters in one of three settings: sales, elections, and social media. In the baseline, the LLM writes the text based on a simple prompt, in the two treatments, the LLM is fine-tuned using either RFT or TFB using the audience signal as feedback. They find that RFT and TFB each generally beat the baseline in terms of head-to-head audience approval (though the comparison between RFT and TFB is inconclusive), and they find that RFT and TFB both result in the LLM-generated text exhibiting more "misaligned" qualities (e.g. misrepresentation or disinformation).

**Strengths:**

S1. The experimental setup is clearly described, and robustness to a few key components is tested (multiple datasets, multiple LLMs, multiple RL algorithms).

S2. The case studies highlighting individual examples are a nice way to illustrate the results presented in Table 2.

**Weaknesses:**

W1. The main weakness of this paper is that it (implicitly) claims to be studying a multi-agent phenomenon (misalignment via competitive pressure), but in practice the experiments appear to only address a single-agent phenomenon (misalignment via reward misspecification). Specifically:
* The main research question is framed around market competition, which suggests the authors are investigating a multi-agent phenomenon. This perspective is further reinforced by using the phrase "Moloch's Bargain" in the title (presumably referencing *Meditations on Moloch*, a blog post on perilous equilibria arising from multi-agent competition, though this connection is not explicitly drawn).
* However, the only sense in which competition is taking place is that a majority vote is used to select the generations to fine-tune on (in the RFT/TFB treatments). The setting otherwise does not resemble economic competition: (1) the learning (training) procedure only involves a single agent (the "competition" is just via multiple rollouts), (2) there is no memory between periods (training steps).
This paper would be stronger if it either was reframed as a study of single-agent reward misspecification, or (more interestingly) if the experiments were redesigned to isolate the misalignment driven specifically by true multi-agent competition.

W2. The authors write (Ln 301-2) "Overall, these findings indicate that textual feedback is a promising approach." However, TFB appears to win against RFT slightly at best (only 4/6 of cases in Table 1, sometimes by slight margins).

W3. The related work does not discuss the literature on novel misalignment risk from multi-agent LLM interactions. One early work on market competition in this space would be the 2023 paper "Algorithmic Collusion by Large Language Models".

W4. I am not particularly convinced by TFB as a training procedure. First, it's unclear whether such a training procedure is particularly operationalizable (we typically lack access to real humans' thoughts). Second, in response to (Ln 240-1) "[t]hese thoughts can identify, for example, which aspects of a sales pitch were compelling and which were not": To what extent is it true that what the customer says they like is the same as what the customer actually likes? The greater the misalignment, the greater the extent to which TFB might be underperforming in the sense of Table 1. Third, in response to (Ln 246-7) "[t]he training objective is then augmented to jointly predict both the trajectory preferred by the majority my and the thoughts ti from all k audience members": This appears to put optimization pressure towards writing pitches such that the audience's thoughts on them are more predictable. This seems like a different kind of interesting unintended effect that perhaps warrants separate exploration.

**Questions:**

Q1. Can the authors explain the extent to which they view their contribution as highlighting multi-agent risks specifically?

Q2. (Minor) What is the sample size for Table 2? Are the slight differences (e.g. +0.10) statistically significant?

---

### Official Review · Reviewer_hAhz · 2025-10-29

**Soundness:** 3
**Presentation:** 4
**Contribution:** 3
**Rating:** 4
**Confidence:** 3

**Summary:**

This paper studies the potential negative consequences of optimizing LLMs for competitive advantage in market environments such as sales, elections, and social media. Using simulated environments and audiences, the study shows that optimizing LLMs for market goals may lead to misalignment, where the models exhibit harmful behaviors such as dishonesty, misinformation, and populist rhetoric. The author refers to this phenomenon as “Moloch’s Bargain,” highlighting the trade-off between performance enhancement and alignment. The paper introduces the simulated environments used for training and evaluating LLMs and compares various methods, such as refusal fine-tuning and text feedback. The findings emphasize the need for stronger governance measures to prevent market-driven dynamics from undermining social trust.

**Strengths:**

1. This paper introduces a novel viewpoint: market-driven optimization may lead to model misalignment, exploring a relatively under-studied area through simulated environments.

2. The paper demonstrates that optimizing LLM performance for critical domain deployment can induce model misalignment. This offers valuable evidence for further research in AI alignment

3. The authors clearly demonstrates how misalignment emerges during optimization, backed by intuitive empirical evidence.

**Weaknesses:**

1. The study is limited to two 8B models (Qwen/Qwen3-8B and meta-llama/Llama-3.1-8B-Instruct). Without validations across more sizes or architectures, the generality of its conclusions is limited.

2. The study hinges on only a single LLM (gpt-4o-mini) to simulate an audience of merely 20 personas. There is no experiment validating whether the LLM-simulated behavior aligns with that of real human.

3. The reproducibility of this work remains questionable. The playgrounds and the code have not been released, nor is there any indication that they will be open-sourced. Furthermore, many experimental details, such as the computational resources (e.g., GPUs) and training cost, are not provided in the current paper.

**Questions:**

1. In the playground, the simulation relies on a single LLM and the audience is limited to only 20 LLM-driven personas. It would be much more informative if the authors could conduct a sensitivity analysis by varying the audience size (k) or the audience model (fp).

2．The paper only evaluates 8B-scale models—does the “Moloch’s Bargain” phenomenon intensify or diminish as models scale up (e.g., to 30B,70B)?

3. The paper presents results across different settings, but lacks sufficient statistical-significance tests. What are the variances of these results?

4. The paper only compares RFT and TFB. What about other RL algorithms, such as GRPO or DPO? Could they yield different optimization outcomes or stronger alignment?

---

### Official Review · Reviewer_25My · 2025-11-01

**Soundness:** 2
**Presentation:** 2
**Contribution:** 2
**Rating:** 2
**Confidence:** 2

**Summary:**

This paper provides empirical evidence of emergent misalignment in LLMs under competitive market incentives, using three open-source simulation environments (sales, elections, social media). It proposes RFT and TFB training methods and quantifies performance-harm trade-offs, framing the phenomenon as “Moloch’s Bargain.”

**Strengths:**

- The paper introduces the “Moloch’s Bargain” framework, which links LLM deceptive behaviors to competitive market pressures, providing a novel lens for understanding emergent misalignment.
- This paper proposes two training algorithms, RFT and TFB to optimize LLM performance in competitive market settings. Experiments show misalignment with these training approaches.

**Weaknesses:**

- In Section 3, the competitive market setting lacks mathematical formalization, which results in unclear objective functions for the subsequent optimization modeling (RFT, TFB).

- The assumption of independent customer decisions in Figure 2 is an unexplained simplification. Group influences are likely to affect customer behavior, which the model does not account for.
- From mathematical optimization perspective, both RFT and TFB can be viewed as variants of supervised fine-tuning (SFT). The paper should further clarify the theoretical rationale for employing the SFT paradigm to optimize agent behavior.

**Questions:**

- The study should justify why GPT-4o-prompted agents can effectively simulate user feedback signals in real-world environments.

---

### Official Review · Reviewer_wBLc · 2025-11-01

**Soundness:** 2
**Presentation:** 3
**Contribution:** 2
**Rating:** 4
**Confidence:** 4

**Summary:**

This paper studies the `Moloch’s Bargain` in LLM persuasion investigating whether training models for improved persuasive ability results in decreased truthfulness. The authors construct synthetic persuasion–counterpersuasion dialogues between fine-tuned GPT-4o-mini agents, trained with text-feedback (RFT) supervision, and claim that increasing persuasiveness induces a measurable truthfulness decline. The work frames this as an emergent trade-off between social influence and epistemic integrity.

**Strengths:**

1. The paper tackles a socially consequential question whether optimizing models for persuasiveness inherently degrades truthfulness and does so with a clean experimental framing (“Moloch’s bargain”).
2. The empirical pipeline is reproducible and systematic, leveraging text-feedback (RFT) as a training mechanism to model persuasive reinforcement.
3. The authors make an effort to quantify persuasion–truth trade-offs across model checkpoints, for examining multi-objective trade-offs,  and study is robust with human evaluation.

**Weaknesses:**

1. The persuasive behavior is entirely measured using LLM audiences. Prior works Singh et al. (2024), Anthropic (2024), and Hackenburg et al. (2024) have shown that LLMs are poor estimators of persuasive efficacy, even at GPT-4o or Claude-class scales. Using GPT-4o-mini as both persuader and judge makes the results difficult to interpret or trust. GPT-4o-mini is too small to meaningfully simulate audience psychology. Evidence in [1-3] indicates that persuasiveness follows a log-scaling law with size; results from sub-10B models tell little about real-world persuasion dynamics.

2. The related work overlooks key persuasion studies [1–3], which already quantify scaling behavior, persuasion–truth correlations, and evaluation reliability. Without situating the contribution relative to these, the work reads as disconnected from current understanding.

3. The truthfulness decline could be an artifact of synthetic feedback loops and hallucination drift, not a genuine persuasion-truth trade-off. Since [1] finds only a modest truthfulness drop even with real-data fine-tuning, the large degradation reported here is likely confounded by simulation noise.

4. No evaluation on human-grounded datasets like CMV or PersuasionArena. Without grounding in real discourse, the measured persuasion gains remain speculative.

**Questions:**

Why was GPT-4o-mini selected for both generation and evaluation, given known weaknesses of smaller models in capturing persuasive or truthful reasoning?

Can the authors clarify how they disambiguate truthfulness degradation from feedback hallucination or reward-model bias in the RFT pipeline?

---

### Note · Authors · 2025-12-13

I have read and agree with the venue's withdrawal policy on behalf of myself and my co-authors.